# Prevalence and correlates of diagnosed and undiagnosed epilepsy and migraine headache among people with severe psychiatric disorders in Ethiopia

**Getinet Ayano**[1,2]*, **Sileshi Demelash**[3], **Zegeye Yohannes**[1], **Kibrom Haile**[1], **Light Tsegay**[4], **Abel Tesfaye**[1,5], **Kelemua Haile**[1], **Mikias Tulu**[1], **Belachew Tsegaye**[1], **Melat Solomon**[1], **Getahun Hibdye**[3], **Dawit Assefa**[1], **Berihun Assefa Dachew**[2]

1 Research and Training Department, Amanuel Mental Specialized Hospital, Addis Ababa, Ethiopia,
2 School of Public Health, Curtin University, Perth, WA, Australia, 3 Ethiopian Public Health Institute, Addis Ababa, Ethiopia, 4 Department of Psychiatric Nursing, College of Health Sciences, Axum University, Axum, Ethiopia, 5 Department of Medicine, Hawassa University, Hawassa, Ethiopia

* babiget2015@gmail.com

**Data Availability Statement:** All relevant data are within the manuscript and its supporting information files.

## Abstract

### Background

There is a paucity of research on the prevalence of diagnosed as well as undiagnosed neurological disorders with episodic manifestations such as epilepsy and migraine headaches in people with severe psychiatric disorders (SPD). To the best of our knowledge, this is the first study analyzing and comparing the prevalence of diagnosed and undiagnosed chronic neurological disorders with episodic manifestations including epilepsy and migraine headache in people with SPD.

### Method

This quantitative cross-sectional survey was undertaken among 309 patients with SPD selected by a systematic random sampling technique. The Structured Clinical Interview for DSM-IV Axis I Disorders (SCID) was used to confirm SPD among the participants. The International Classification of Headache Disorders (ICHD-3) and International League Against Epilepsy (ILAE) were used to define migraine headache and epilepsy, respectively]. Risk factors for chronic neurologic disorders were explored by using logistic regression models.

### Result

In this study, the prevalence of overall neurological disorders, epilepsy, and migraine headache among people with SPD were found to be 5.2% (95%CI 3.2–8.3), 1.6% (95%CI 0.7–3.9), and 3.9% (95%CI 2.2–6.7), respectively. We found that a considerably higher proportion of people with SPD had undiagnosed overall neurological disorder (87.5%; 14/16), epilepsy (60%; 3/5), as well as migraine headaches (100%; 12/12). On the other hand, in this study, 12.5%, 40%, and 0% of patients with overall neurologic disorder, epilepsy, and

**Funding:** This research work is funded by Amanuel mental Specialized Hospital, Addis, Ababa, Ethiopia.

**Competing interests:** The authors have declared that no competing interests exist.

migraine headaches respectively were diagnosed by the professionals. Higher disability score (WHODAS score) was associated with increased odds of having neurological disorders compared with the lower WHODAS score [OR = 1.30 (95% CI 1.02–1.66)].

## Conclusion

Whilst the prevalence estimates of neurological disorders with episodic manifestations including epilepsy and migraine headache was high among people with SPD, the vast majority of them remained undiagnosed. The diagnosis rates of those disorders were significantly low, perhaps surprisingly zero for migraine headache. High WHODAS score was associated with increased odds of having neurological disorders. Routine screening and management of epilepsy and migraine headache are imperative among people with SPD.

## Background

Severe psychiatric disorders (SPD) most commonly referring to the diagnosis of mental disorders with a substantial impairment over multiple domains [1]. These include schizophrenia, bipolar, schizoaffective, and depressive disorder [1]. According to the global burden of disease (GBD), SPDs are key contributors to the global burden of disease and they are among the major causes of morbidity and mortality worldwide [2, 3]. The prevalence estimates of SPD in adult ranges from 4 to 6%, which shows that a notably higher proportion of adults have SPD [4, 5].

Research evidence shows that a considerable percentage of people with SPD have comorbid medical conditions including neurologic disorders such as epilepsy and migraine headaches [6–9]. For example, a recent meta-analysis that assessed the prevalence of migraine headaches among patients with bipolar disorder involving seven studies on the subject found that roughly one-third of patients with bipolar disorder had comorbid migraine headaches (30.36%) [8]. However, the existing literature indicates that: (1) the vast majority of co-occurring medical conditions were untreated and undertreated; (2) most of them remain undiagnosed [10]; (3) even among those patients receiving treatment, the quality of service/care for both the comorbid medical condition and psychiatric disorders has been inadequate [11, 12]. Scientific evidence shows that early identification and treatment of comorbid medication conditions in patients with severe psychiatric disorders are associated with improvement in outcomes, increased quality of life, reduced burden and cost associated with medical services, and improved in functionality/productivity [13, 14].

Even though there are no previous studies that reported the rates of undiagnosed neurologic disorders with episodic manifestation (such as epilepsy and migraine headaches) among patients with SPD, evidence from the general population shows that the vast majority of patients with epilepsy and migraine headache remained undiagnosed in the general community. For example, in a recent study that assessed the prevalence of undiagnosed migraine headaches using 15000 US households selected by a door-to-door survey, the prevalence of undiagnosed migraine was 70% for males and 60% for females [15].

While there is a paucity of supportive epidemiologic data on the prevalence estimate of undiagnosed neurologic disorder (epilepsy and migraine headache) among people with SPD, it is assumed that the prevalence rates of epilepsy and migraine headache believed to be significantly higher in patients with SPD as compared to reported prevalence rates from the general population due to: (1) both SPD and neurologic disorders have common clinical

manifestations which might increase the rates of undiagnosed disorders; (2) the nature of psychiatric disorders (SPD) which remarkably impairs the understanding and insight of the patients [16, 17]; (3) the reported higher prevalence rates of undiagnosed physical illness including epilepsy and headaches (up to 80%) in people with mental illness [7, 18], suggesting the high probability of those neurologic disorders to be undiagnosed.

To the best of our knowledge, this is the first study analyzing and comparing the prevalence of diagnosed and undiagnosed chronic neurologic disorders with episodic manifestations including epilepsy and migraine headache in people with SPD. The results will help to understand the approximate pictures of undiagnosed disorders (epilepsy and migraine headache) among patients with SPD and to suggest potential clinical and research recommendations.

## Methods

### Study design and period

This quantitative cross-sectional study was conducted among 309 patients with severe psychiatric disorders from May 1, 2017, to July 30, 2017. All participants were recruited from Amanuel Mental Specialized Hospital, Addis Ababa, Ethiopia, which is the only specialized psychiatry hospital in the country.

### Inclusion and exclusion criteria

Participants satisfying the following criteria were included in this study: (1) adult (aged 18 years and above); (2) those patients who had a diagnosis SPD (including schizophrenia, bipolar, schizoaffective, and depressive disorders). Those patients who had a severe illness at the time of data collection (affecting communications) were excluded.

### Sampling procedure

This survey is part of a comorbidity study among patients with severe psychiatric disorders in a specialized mental health setting in central Ethiopia. Hence, the sample size has been calculated considering the prevalence of comorbid physical conditions among people with SPD nearly 80% [7, 19]. The following assumptions were considered to calculate the sample size: (a) 95% confidence interval; (b) 80% proportion of comorbid medical conditions; (c) 30% non-response rate. The final sample size was 320.

Participants were selected using a systematic random sampling technique. The sampling interval (K = 11) was identified by dividing the total participants with SPD who had the follow-up and treatment during the study period by total sample size. A lottery method was used to select the first study participant, and a regular interval was used to select the remaining participants.

## Measures

### Diagnosis of severe psychiatric disorders

The structured clinical interview for DSM- Ⅴ-TR axis  disorders (SCID) was used to ascertain the presence of severe psychiatric disorders [20]. SCID is a validated instrument designed to assess the diagnosis of the Diagnostic Manual of Mental Disorders Axis I disorders (DSM-IV) (major psychiatric disorders), which is extensively used to assess those disorders in previous studies conducted in Ethiopia [21, 22].

## Diagnosis of epilepsy

The current diagnosis of Epilepsy was identified according to the International League Against Epilepsy (ILAE) classification of seizure criteria, a neurological condition characterized by two or more unprovoked seizures that occurred at least 24 hours apart [23]. According to the ILAE guideline, an epilepsy case is defined as someone with an active, recurrent (two or more) condition of an epileptic seizure, which was unprovoked by an immediate cause. The same classification and definition of epilepsy have been used in previous studies in Ethiopia [24, 25]. The previous diagnosis of epilepsy was taken from the chart of the patients (yes/no).

## Migraine headache case definition

The present case of migraine headache was defined based on the International Classification of Headache Disorders (ICHD-3) criteria of migraine headaches with or without aura [26].

The previous diagnosis of migraine was taken from the chart of the patients (yes/no).

## Screening for disability

In this study, disability was assessed using the World Health Organization Disability Assessment Schedule (WHODAS-2), an instrument designed to measure functional impairment [27]. This tool covers six domains of functioning including cognition, mobility, self-care, getting along (interacting with other people), life activities, and community participation (25) and has been validated in Ethiopia on patients with severe psychiatric disorders [28].

## Sociodemographic and other factors

We used structured questionnaires to collect data on sociodemographic and clinical characteristics such as age, marital status, educational status, ethnicity, religion sex, residence, suicide, duration of the illness, history of hospitalizations, and relapse. Trained psychiatry professionals, who have adequate knowledge and experience on SCID criteria, collected data.

**Definitions of terms.** In the present study, overall neurological disorders represent the presence of either epilepsy and migraine headaches in patients with SPD. Research studies suggest that epilepsy and migraine headaches are the two commonest chronic neurological disorders with episodic manifestations [29].

**Data quality control.** In this study, structured and interviewer-administered questionnaires that have been developed in English were translated into the local language (Amharic), and then to check the consistency the Amharic version of the questionnaires was translated back into English. Assessors with adequate knowledge and experience about the diagnostic statistical Manual of Mental Disorders (DSM), ICHD-3, and ILAE were recruited (MSc psychiatry professionals) to assure the quality of the data. Moreover, adequate training has been given regarding sampling procedure or protocol, on how to complete the questionnaire, eligibility criteria, ethical issues, data collection procedure, and the components of the questionnaire (sociodemographic and other variables), as well as the details of the two main data collection instruments (ICHD-3 and ILAE) for the supervisors and data collectors. The questionnaires have been pretested before the actual data collection and the essential modification was undertaken. Two Supervisors followed the data collection process and the necessary correction was made when needed. The supervisor and principal investigator reviewed the collected data and checked for completeness and relevance each day.

## Statistical analysis

Stata (version 16) was used to conduct all the statistical analysis. Descriptive statistics for the participants were provided. Frequency/percentage was used to express categorical variables

and mean (standards deviations) were used for continuous variables. The dependent variables (diagnosed and undiagnosed cases of neurologic disorders i.e. epilepsy and migraine headache) were measured using percentage. We conducted bivariate and multivariate logistic regression analysis to look at the association between dependent and independent variables. OR with 95% CI was used to measure the strength of the association and a P-value less than 0.05 was considered as statistically significant.

## Ethical consideration

The study protocol was approved by Amanuel Mental Specialized Hospital (Research and training department) ─the human research and ethics committee (HREC). Informed written consent has been obtained from every participant after a clear and detailed explanation of the purpose, significance, objectives, harms, and benefits of participation.

## Results

### Sociodemographic characteristics of the participants

In this study, a total of 320 patients has been assessed for eligibility, out of which 309 (96.6%) patients with SPD including schizophrenia (n = = 135), bipolar (n = 54), schizoaffective (n = 28), and depressive (n = 92) disorders were included in the final analysis. The mean (±SD) age of the participants was 36.19 (± 10.45) years. Most of the participants were male (65.4%) and single by marital status (65.4%). More than one-third of the participants had attended secondary school, approximately half of the participants were Orthodox Christians (51.47%), and more than three-fourth were from urban areas (76.05%) (Table 1).

### The prevalence of neurologic disorders in patients with SPD

The overall prevalence of neurologic disorders (including epilepsy and migraine headache combined) in our study was 5.2% (95%CI 3.2–8.3). Specifically, the prevalence was 1.6% (95% CI 0.7–3.9)) for epilepsy and 3.9% (95%CI 2.2–6.7) for migraine headaches.

Additionally, the prevalence of comorbid migraine headache among epileptic patients was 20%, whereas the prevalence of comorbid epilepsy among migraine headache patients was 8.3%. We also found that 0.3% of patients with SPD had both epilepsy and migraine headache.

### The prevalence of diagnosed and undiagnosed neurological disorders in patients with SPD

The prevalence of undiagnosed neurological disorders was 4.5% among the total participants and 87.5% of those participants with neurological disorders. Likewise, the prevalence of undiagnosed epilepsy was 1% among the total patients and 60% in patients with epilepsy.

Regarding migraine headache, the prevalence of undiagnosed migraine headache was 3.9% of the total participants and 100% in patients with migraine headaches (Table 2).

On the other hand, in this study, 12.5%, 40%, and 0% of patients with overall neurologic disorder, epilepsy, and migraine headaches respectively were diagnosed by the professionals.

### Factors associated with undiagnosed neurological disorders in patients with SPD

Multivariable logistic regression analysis revealed that none of the socio-demographic and clinical characteristics of participants included in the model were found to be associated with undiagnosed neurological disorders in patients with SPD. Higher WHODAS score, however,

**Table 1. Sociodemographic characteristics of the participants with severe psychiatric disorders in Addis Ababa, Ethiopia (*n* = 309).**

| Characteristics | Frequency | Percentage |
|---|---|---|
| **Sex** | | |
| Male | 202 | 65.37 |
| Female | 107 | 34.63 |
| **Age** | | |
| 30 or less | 110 | 35.6 |
| 30 to 40 | 106 | 34.3 |
| 41 and more | 93 | 30.1 |
| **Educational status** | | |
| Uneducated | 30 | 9.71 |
| Primary | 103 | 33.33 |
| Secondary | 118 | 38.19 |
| Higher | 58 | 18.77 |
| **Religion** | | |
| Muslim | 87 | 28.16 |
| Orthodox | 157 | 51.46 |
| Protestant | 57 | 18.54 |
| Others | 6 | 1.94 |
| **Marital status** | | |
| Single | 202 | 65.37 |
| Married | 74 | 23.95 |
| Divorcee/widowed | 33 | 10.68 |
| Ethnicity | | |
| Amhara | 95 | 30.74 |
| Oromo | 91 | 29.45 |
| Gurage | 82 | 26.54 |
| Others | 41 | 13.27 |
| Residence | | |
| Urban | 235 | 76.05 |
| Rural | 74 | 29.95 |
| **SPD type** | | |
| Schizophrenia | 135 | 43.69 |
| Bipolar disorder | 92 | 29.77 |
| Depressive disorder | 54 | 17.48 |
| Schizoaffective disorders | 28 | 9.06 |
| **Catatonia** | | |
| No catatonia | 248 | 80.26 |
| Catatonia | 61 | 19.74 |
| **Psychosis** | | |
| No psychosis | 44 | 14.24 |
| psychosis | 265 | 85.76 |
| **History of relapse** | | |
| Relapsed | 226 | 73.14 |
| No relapse | 83 | 26.86 |
| **History of admission** | | |
| Admission | 196 | 63.42 |
| No admission | 113 | 36.57 |

**Table 2. The prevalence of undiagnosed neurologic disorders among patients with severe mental disorders in central Ethiopia, n = 309.**

| Disorder | Chart diagnosis, n (%) | Real diagnosis, n (%) | Undiagnosed disorder from the total, n (%) | Undiagnosed disorder from the cases, n (%) |
|---|---|---|---|---|
| Overall neurologic disorders | 2 (0.65) | 16 (5.18) | 14 (4.53) | 14 (87.50) |
| Epilepsy | 2 (0.65) | 5 (1.62) | 3 (0.97%) | 3 (60.00) |
| Migraine headache | 0 (0.00) | 12 (3.88) | 12 (3.88) | 12 (100) |

**Key**: Chart diagnosis indicates the diagnosis of the patient taken from the chart, while real diagnosis indicates the current diagnosis of the patient according to the assessors (tools used).

was associated with increased odds of having neurological disorders when compared with the lower WHODAS score [OR = 1.30 (95% CI 1.02–1.66)] (Table 3).

# Discussion

## Main findings

To the best of our knowledge, this is the first study that examined the epidemiology of diagnosed and undiagnosed neurological disorders, including epilepsy and migraine headache in patients with severe psychiatric disorders in a specialized psychiatric setting. Our results indicated that a considerable proportion of people with severe psychiatric disorders had diagnosed and undiagnosed neurological disorders. The highest prevalence estimate of undiagnosed disorder was observed for migraine headache (100%) followed by overall neurological disorders (87.5%) and epilepsy (60%). Our findings suggest that routine screening and intervention for neurological disorders should be considered in people with SPD.

## The prevalence and associated factors of neurological disorders in people with SPD

The prevalence of neurological disorders including epilepsy and migraine headache in this study were found to be much higher than the reported global prevalence of those disorders in the general population [24, 30–35]. For instance, a recent meta-analysis including 197 prevalence studies conducted across the globe revealed that the lifetime prevalence of epilepsy in the general community was 7.6 per 1000 population [36], which is considerably lower than the reported prevalence epilepsy among patients with severe psychiatric disorders(16.2 per 1000 population) in the current study. A recent meta-analysis conducted in Ethiopia reported that the prevalence of epilepsy in the general community was 5.2 per 1000 population) [35]. Our result was 3.12 times higher than this reported prevalence. There are wide ranges of explanations for higher prevalence rates of epilepsy among people with severe psychiatric disorders. One of the possible explanations is that these disorders might have shared genetic factors. For example, a recent genetic study by Lopez et.al found that epilepsy and bipolar disorders have a common genetic abnormality [37]. The above study revealed that abnormalities in ANK3-coded proteins in the brain (lower amount of ANK3 type proteins) in these disorders, which is responsible for increasing output (excitation) and holding back out (inhibitions) [37]. Additionally, epidemiologic evidence demonstrates that these two disorders have been treated by the same drugs including sodium valproate (VPA) and carbamazepine (CBZ), which are effective drugs for both disorders, indicating some underlying common pathways across these disorders [38]. Similarly, a genetic-based study conducted in the USA revealed that epilepsy and schizophrenia have a shared genetic abnormality including abnormal development of the

**Table 3. Factors associated with neurologic disorders in people with severe mental disorders, Addis Ababa, Ethiopia.**

| Characteristics | Neurologic disorders | | Crude odds ratio (95%CI) | Adjusted odds ratio (95%CI) |
|---|---|---|---|---|
| | Yes | No | | |
| **Gender** | | | | |
| Female | 6 | 101 | 1.14 (0.40–3.23) | 1.01 (0.33–3.08) |
| Male | 10 | 192 | 1 | 1 |
| **Age** | | | | |
| <35 | 6 | 135 | 1 | 1 |
| ≥ 35 | 10 | 158 | 1.42 (0.50–4.02) | 1.34(0.34–3.35) |
| **Residence** | | | | |
| Urban | 15 | 220 | 1 | 1 |
| Rural | 1 | 73 | 0.20 (0.26–1.55) | 0.19 (0.02–1.60) |
| **Marital status** | | | | |
| Single | 10 | 192 | 1 | 1 |
| Married | 3 | 71 | 0.81 (0.21–3.03) | 1.07 (0.27–4.28) |
| Divorce/widowed | 3 | 30 | 1.92 (0.50–7.34) | 2.37(0.53–10.59) |
| **Catatonia** | | | | |
| No catatonia | 11 | 237 | 1 | 1 |
| Catatonia | 5 | 56 | 2.56 (0.64–5.76) | 1.85 (0.59–5.87) |
| **Psychosis** | | | | |
| No psychosis | 2 | 14 | 1 | 1 |
| Psychosis | 14 | 251 | 1.17 (0.26–5.34) | 1.07 (0.21–5.53) |
| **Misdiagnosed severe psychiatric disorder** | | | | |
| Correct diagnosis | 13 | 175 | 1 | 1 |
| Misdiagnosis | 3 | 118 | 0.34 (0.09–1.23) | 0.31 (0.08–1.15) |
| **Relapse** | | | | |
| Relapsed | 13 | 23 | 1.63 (0.45–5.86) | 1.38 (0.35–5.47) |
| No relapse | 3 | 80 | 1 | 1 |
| **Admission** | | | | |
| Admission | 11 | 185 | 1.28 (0.43–3.79) | 1.03(0.31–3.42) |
| No admission | 5 | 108 | 1 | 1 |
| WHODAS score | | | 1.21 (1.00–1.51) | **1.30 (1.02–1.66)**∗ |

∗ Significant association (p-value < 0.05).

brain and nervous system [39, 40]. Likewise, the existing scientific evidence indicates that depression and epilepsy have a shared genetic abnormity responsible for causing these disorders [41]. The other possible explanation for the higher prevalence rates of epilepsy among people with severe psychiatric disorders is having underlying common neurotransmitter abnormalities (neurochemical underpinnings) [42].

Regarding the prevalence of migraine headache, although we did not found previous studies concerning migraine headache among patients with overall severe psychiatric disorders (including schizophrenia, bipolar, schizoaffective, and depressive disorders combined), there are several studies conducted in specific categories of disorders. For example, a recent meta-analysis of seven studies on the subject showed that the prevalence of migraine headache among patients with bipolar disorders was 30.36% [8], which was significantly higher than the reported prevalence in the current study on severe mental disorders (3.88%). Another study that assessed the prevalence of migraine headache among schizophrenia patients found a remarkably lower prevalence (2%) [43] when compared with the prevalence in this study. The

prevalence of migraine headache in the current study was more than 2 times higher than the reported prevalence in the general population, according to the global burden of disease study in 2016 (1.8%) [44]. The highest prevalence of migraine headache among patients with severe psychiatric disorders could be due to common genetic as well as environmental factors responsible for both migraine headache and severe psychiatric disorders [45, 46].

As for the associated factors, in this study, disability was associated with the diagnosis of neurological disorders. After adjusting all the potential confounders, greater disability (the highest WHODAS score) was associated with increased odds of having neurological disorders when compared with lower disability (the lower WHODAS score) [OR = 1.30 (95% CI 1.02–1.66)]. Consistent with our finding a recent study found a significant and positive association between greater disability and increased risks of depression [47].

## The prevalence of undiagnosed neurological disorders diseases in people with SPD

In the current study, approximately nine out of ten (87.5%) participants with neurologic disorders were undiagnosed among people with SPD. For specific disorders, the rate of undiagnosed disorder was relatively higher for migraine headaches (100%) followed by epilepsy (60%), which were remarkably higher than the reported prevalence rates in the general population. For example, in a recent study that assessed the prevalence of undiagnosed migraine headache in the US revealed that the prevalence of undiagnosed migraine was 70% for males and 60% for females [15], which is remarkably lower than the reported magnitude in the current study (100%).

The possible reason for higher rates of undiagnosed neurological disorders (migraine headache and epilepsy) might be due to a significant overlap between symptoms of severe psychiatric disorder and neurological disorders. For instance, a study conducted by Moeno et.al found that more than one fourth (25.4%) of patients with major depressive disorders visited a primary care setting with a chief complaint of headache [48]. Similarly, a case-control study conducted by Marlow et.al reported that 32% of patients who reported headaches as the main symptoms were diagnosed with major depressive disorders [49]. More recently, a cross-sectional study that evaluated headache and schizophrenia revealed that 57% of patients with schizophrenia had an overlapping headache as a clinical presentation [50]. More recently, a cross-sectional study that evaluated headache and schizophrenia revealed that 57% of patients with schizophrenia had overlapping headaches as a clinical presentation [51, 52].

Likewise, a significant proportion of patients with epilepsy have overlapping psychiatric symptoms [53, 54], which might partly explain the higher levels of undiagnosed epileptic cases in patients with severe psychiatric disorders. The other possible reason for the observed higher rates of undiagnosed neurological cases could be due to the severity of the psychiatric disorders, which impairs the insight of the patients to adequately understand and report their complaints. Complementing this, a cross-sectional study that has assessed insight in patients with schizophrenia found that 50–60% of patients lack insight (either partially or completely) into their mental disorders [55]. Similarly, epidemiologic data shows that a significant proportion of patients with depressive and bipolar disorders have a lack of insight into their disorder [56, 57]. The skills, training as well as knowledge of the psychiatry professionals about those neurological disorders might be the other possible reasons for higher prevalence rates of undiagnosed disorders.

## Recommendations for future research and clinical practice

This study has some implications for future clinical practice and research: (1) we found that rates of undiagnosed neurological disorders are significantly high but the estimates for the

distinct categories SPD are not explored because of a small number of study participants for the specific disorder, indicating the need for future studies addressing these gaps. (2) This study found a higher prevalence of undiagnosed neurological disorders when compared with the reported prevalence in the general population, which needs future robust studies regarding the potential reasons for the discrepancies. (3) Training and strategies (including continuous medical education (CME)) to increase awareness of the neurological disorders, as well as the misdiagnosis rates for the psychiatric professionals, is warranted. (4) Routine screening and intervention for neurological disorders should be considered in people with SPD. (5) Future studies on the underlying factors for the higher rates of misdiagnosis are needed.

## Strengths and limitations

This study has several strengths to note: (1) It is the first study analyzing and comparing the prevalence estimates of undiagnosed neurological disorders among people with SPD; (2) severe psychiatric disorders were confirmed by a standard and diagnostic instrument (SCID). (3) We have used a standard instrument to ascertain severe psychiatric disorders. (4) Epilepsy and migraine headache were defined based on standard measuring tools. However, this study has some potential shortcomings that should be noted: First, factors associated with neurological disorders may not imply causality as the study design was cross-sectional. Secondly, the use of chart records for diagnosis of neurological disorders might underestimate the diagnosed cases since some previous diagnoses may not be documented. Thirdly, the small number of participants (sample) should be considered in interpreting the results especially for the estimates of the specific disorders such as epilepsy and migraine headache. Also, future studies addressing this limitation are warranted.

## Conclusion

In summary, this study showed that a significant proportion of people with SPD had neurological disorders including epilepsy and migraine headache. The highest prevalence estimate of undiagnosed disorder was observed for migraine headache followed by overall neurological disorders and epilepsy. Our findings suggest that routine screening and intervention for neurological disorders should be considered in people with SPD. Training and strategies to increase awareness of neurological disorders, as well as the misdiagnosis rates for psychiatric professionals, are warranted. Future longitudinal studies with adequate sample size are needed to identify the factors associated with undiagnosed neurological disorders particularly focusing on the specific disorders.

## Supporting information

**S1 File.**
(XLS)

## Acknowledgments

We are very grateful to the study participants for their cooperation in providing the necessary information and giving us their precious time.

## Author Contributions

**Conceptualization:** Getinet Ayano.

**Data curation:** Getinet Ayano, Zegeye Yohannes, Kibrom Haile, Mikias Tulu, Melat Solomon, Getahun Hibdye.

**Formal analysis:** Getinet Ayano.

**Funding acquisition:** Getinet Ayano.

**Investigation:** Getinet Ayano, Sileshi Demelash, Zegeye Yohannes, Kibrom Haile, Light Tsegay, Abel Tesfaye, Kelemua Haile, Mikias Tulu, Belachew Tsegaye, Melat Solomon, Getahun Hibdye, Dawit Assefa, Berihun Assefa Dachew.

**Methodology:** Getinet Ayano, Sileshi Demelash, Kibrom Haile, Light Tsegay, Abel Tesfaye, Kelemua Haile, Mikias Tulu, Belachew Tsegaye, Melat Solomon, Getahun Hibdye, Dawit Assefa, Berihun Assefa Dachew.

**Project administration:** Getinet Ayano, Mikias Tulu.

**Resources:** Getinet Ayano, Zegeye Yohannes, Kibrom Haile.

**Software:** Getinet Ayano.

**Supervision:** Getinet Ayano, Zegeye Yohannes, Kibrom Haile, Kelemua Haile, Mikias Tulu, Belachew Tsegaye, Dawit Assefa.

**Validation:** Getinet Ayano, Mikias Tulu, Berihun Assefa Dachew.

**Visualization:** Getinet Ayano, Kibrom Haile, Abel Tesfaye.

**Writing – original draft:** Getinet Ayano.

**Writing – review & editing:** Getinet Ayano, Sileshi Demelash, Zegeye Yohannes, Kibrom Haile, Light Tsegay, Abel Tesfaye, Kelemua Haile, Mikias Tulu, Belachew Tsegaye, Melat Solomon, Getahun Hibdye, Dawit Assefa, Berihun Assefa Dachew.

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
