## [Decision Letter · Decision Letter 0]

18 Aug 2020

PONE-D-20-11101

The epidemiology of diagnosed and undiagnosed chronic neurological disorders with episodic manifestations in People with Severe Psychiatric Disorders in Ethiopia

PLOS ONE

Dear Dr. Ayano,

Thank you for submitting your manuscript to PLOS ONE. After careful consideration, we feel that it has merit but does not fully meet PLOS ONE’s publication criteria as it currently stands. Therefore, we invite you to submit a revised version of the manuscript that addresses the points raised during the review process.

We look forward to receiving your revised manuscript.

Kind regards,

Kyoung-Sae Na, M.D.

Academic Editor

PLOS ONE

Journal Requirements:

3. We note you have included a table to which you do not refer in the text of your manuscript.

Please ensure that you refer to Table 3 in your text; if accepted, production will need this reference to link the reader to the Table.

Reviewers' comments:

Reviewer's Responses to Questions

**Comments to the Author**

1. Is the manuscript technically sound, and do the data support the conclusions?

Reviewer #1: Yes

Reviewer #2: Partly

Reviewer #3: Yes

2. Has the statistical analysis been performed appropriately and rigorously? 

Reviewer #1: Yes

Reviewer #2: Yes

Reviewer #3: Yes

3. Have the authors made all data underlying the findings in their manuscript fully available?

Reviewer #1: Yes

Reviewer #2: Yes

Reviewer #3: Yes

4. Is the manuscript presented in an intelligible fashion and written in standard English?

Reviewer #1: Yes

Reviewer #2: Yes

Reviewer #3: Yes

5. Review Comments to the Author

Reviewer #1: Also attached.

I find this to a very interesting paper and see no major concern. But, I would like the following points revised.

This paper mainly focussed on co-morbid epilepsy and migraine headache. But, the way it is presented in the title, described in the texts and discussed makes it look like, the focus is many, if not all neurological disorders. Taking this as “Epidemiology of…” also takes this work out of its context. I would suggest this is revisited. “Diagnosed and undiagnosed epilepsy and migraine headache co-morbidity in people with Severe Psychiatric Disorders in Ethiopia” or so.

This is a cross-sectional study where researchers assessed patients or interviewed. My understanding is that all patients were interviewed and I did not see why they looked into past diagnoses. Is this a life time prevalence data? They may need to comment on why the did that.

Readers may like to know more about how they used the instruments. Was it translated, validated? This includes ILAE and ICHD-3. Was this self reported on interview based. Please give some comments.

There were only 3/309 SPD patients who had undiagnosed epilepsy and 12/309 SPD patients with undiagnosed migraine headache. The way it looks in the abstract and texts is a bit exaggerated as 60% and 100%. I suggest that is made clear: 2/3 (60 percent) of co-morbid epilepsy and 12/12 (100 per cent) of co-morbid migraine headache were undiagnosed.

Reviewer #2: Comments for the manuscript titled “The epidemiology of diagnosed and undiagnosed chronic neurological disorders with episodic manifestations in People with Severe Psychiatric Disorders in Ethiopia”.

General comments

Over all, study was done in an important and un addressed area and authors also reported good findings. But major concern what authors should consider is that, most of the patients with severe mental illness have manifestations which are really similar symptoms of neurologic disorders, so how could authors really overcome with his issue and in my suggestion most of the reports you found might be primary symptoms of SPD not neurologic disorders/what are the contents of the tool you used to measure Neurologic disorders like epilepsy in terms commonly reported neurologic like symptoms of SPD?

Section comments

Abstract: your back ground statement and finding are not consistent, I think your focus of study is Epilepsy and migraine headache, but you reported the prevalence of overall neurologic disorder. What was the tool you used to assess over all neurologic disorder, since neurologic disorder includes plenty of disorders? Please clarify what is meaning of overall neurologic disorder?

Your conclusion is not also consistent with your title, what is importance of saying epidemiology of diagnosed and undiagnosed neurologic disorders? Since you only reported the finding of un diagnosed neurologic disorders in your result section of abstract.

You also reported that “Higher disability score (WHODAS score) was associated with increased odds of having neurological disorders” what kind of disability is it? We expect all most all of patients with severe mental disorders have disability, so how can you compare patients with SPD as disable and able? If so in terms of which domain of disability functioning?

Background:

Some modification in grammar, punctuation and English language should be done, for example line number 13 of background section; “Scientific evidence shows that early identification and treatment of comorbid medication conditions in patients with severe psychiatric disorders…….” Should be modified.

Line number 15 “improved” in should be modified as improvement in……

Even though, you reported scarcity of reports regarding prevalence of neurologic disorders among patients with severe mental illness it is better to include different magnitudes and burdens of neurologic disorders particularly epilepsy among patients with mental illness since there are literatures too.

Method:

Study was conducted on 309……. it is better to be modified as study was conducted among 309…..

Inclusion criteria were…… modify also

Exclusion criteria: Your exclusion criteria was patients having severe illness at time of data collection affecting communication, we expect most of patients are with severe illness as name indicates SPD and communication problem is also one of the hallmark symptom of severe mental illness, of so how could you managed this issue.

Sampling procedure: It is good to using systematic random sampling, but my concern is how you managed or calculated k value for more than one treatment units? Since it is expected that more than one OPDs might be available in hospital?

Measures: What was importance of using structured clinical interview for DSM- ΙѴ-TR axis Ι disorders (SCID) since you included already diagnosed patients?

Result: More than one-third of the participants had attended secondary school approximately half of the participants were Orthodox Christians…it needs coma in between school and approximately.

What was prevalence of diagnosed neurologic disorders, since I cannot accesses within your manuscript?

Factors associated with undiagnosed neurological disorders in patients with SPD

You found only higher WHODAS score was only variable associated with prevalence of neurologic disorders, would you explain your variable selection method and what was reason behind for this?

Why don’t authors perform regression separately for epilepsy and migraine headache?

Discussion: Our prevalence was more than 2 times higher than reported prevalence of migraine headache in the general population as reported by the global burden of disease (GBD) study in 2016 (1.8%)……. Modify this sentence

Even though you did not compare with similar studies, your discussion is well written.

Conclusion: Future longitudinal studies are needed to identify the factors associated with undiagnosed neurological disorders (what was difficulty you faced to do this study design)?

Table 3: I strongly recommend you to write cross tabulation results for each variable.

Reviewer #3: Sample size calculation

On sample size calculation, it is not clear that there are many comorbidities with severe psychiatric disorder. Therefore, better to deal with the prevalence of epilepsy and migraine headache. Report that you have considered the maximum sample size.

On statistical analysis

The researcher mentioned the independent variables; just remove from the statistical analysis.

Result

Sociodemographic characteristics of the participants

Rather than mentioning as the following “Sociodemographic and clinical data for our participants were summarized in Table 1” it is better to write (table 1) at the end of the paragraph.

The prevalence of undiagnosed neurological disorders in patients with SPD

Don’t put table 2 in each paragraph. Just put it at the end of last paragraph.

The prevalence and associated factors of neurological disorders in people with SPD

Discussion

The researchers should at least search for additional literature that has been done from different data bases. If no they can compare their finding which is cross-sectional study with the meta-analysis.

Some of the justification for the difference in the finding are not scientifically sound according to the context that they are studying. Which the stating as follows

“There are wide ranges of explanations for higher prevalence rates of epilepsy among people with severe psychiatric disorders as compared with the reported prevalence in the general population. One of the possible explanations is that these disorders might have shared genetic factors”

Therefore, better to explain in more acceptable and in details so that the reader will be convinced from the finding.

References

Outdates references should be corrected by the recently published works. I encourage the researchers even to look into additional references from Ethiopia.

General comment

The manuscript is scientifically sound and well written

6. PLOS authors have the option to publish the peer review history of their article (what does this mean?). If published, this will include your full peer review and any attached files.

Reviewer #1: No

Reviewer #2: **Yes: **Alemayehu Molla

Reviewer #3: **Yes: **Abraham Tamirat Gizaw

---

## [Author Response · Author response to Decision Letter 0]

10 Sep 2020

Point by point responses for reviewers comments

Dear Kyoung-Sae Na, M.D, Academic Editor, PLOS ONE

Thank you very much for giving us the opportunity to rerevise our manuscript “The epidemiology of diagnosed and undiagnosed chronic neurological disorders with episodic manifestations in People with Severe Psychiatric Disorders in Ethiopia”. 

We would also like to thank the reviewers for detailed reviews and providing us with helpful suggestions that will strengthen our manuscript and knowledge. We have gone through the comments and tried to include the responses to all the comments and suggestions. 

Comments by editor and reviewers

 Reviewer #1

Q1. I find this to a very interesting paper and see no major concern. But, I would like the following points revised.

This paper mainly focussed on co-morbid epilepsy and migraine headache. But, the way it is presented in the title, described in the texts, and discussed makes it look like, the focus is many, if not all neurological disorders. Taking this as “Epidemiology of…” also takes this work out of its context. I would suggest this is revisited. “Diagnosed and undiagnosed epilepsy and migraine headache co-morbidity in people with Severe Psychiatric Disorders in Ethiopia” or so.

We addressed this comment accordingly. As the reviewer mentioned the current study did not include all the neurologic disorders. However, as we mentioned in the title as well as throughout the documents the paper addresses the epidemiology (prevalence, detection rates, as well as associated factors) of the two most common neurologic disorders with episodic manifestations such as epilepsy and migraine headache. 

We now revised the title as “Prevalence and correlates of diagnosed and undiagnosed epilepsy and migraine headache among people with Severe Psychiatric Disorders in Ethiopia” 

Q2. This is a cross-sectional study where researchers assessed patients or interviewed. My understanding is that all patients were interviewed and I did not see why they looked into past diagnoses. Is this a life time prevalence data? They may need to comment on why the did that.

The existing diagnoses have been taken from the chart in order to determine the detection rates or the rates of undiagnosed disorders, which require the prevalence of both the current and the existing diagnoses. 

Q3.Readers may like to know more about how they used the instruments. Was it translated, validated? This includes ILAE and ICHD-3. Was this self reported on interview-based. Please give some comments.

We addressed this concern accordingly. We have included the following paragraph in the method sections” 

Data quality control

In this study, structured and interviewer-administered questionnaires that have been developed in English were translated into the local language (Amharic), and then to check the consistency the Amharic version of the questionnaires was translated back into English. Assessors with adequate knowledge and experience about the diagnostic statistical Manual of Mental Disorders (DSM), ICHD-3, and ILAE were recruited (MSc psychiatry professionals) to assure the quality of the data. Moreover, adequate training has been given regarding sampling procedure or protocol, on how to complete the questionnaire, eligibility criteria, ethical issues, data collection procedure, and the components of the questionnaire (sociodemographic and other variables), as well as the details of the two main data collection instruments (ICHD-3 and ILAE) for the supervisors and data collectors. The questionnaires have been pretested before the actual data collection and the essential modification was undertaken. Two Supervisors followed the data collection process and the necessary correction was made when needed. The supervisor and principal investigator reviewed the collected data and checked for completeness and relevance each day”. (See method section page 6-7 line 175-190). 

Q4. There were only 3/309 SPD patients who had undiagnosed epilepsy and 12/309 SPD patients with undiagnosed migraine headaches. The way it looks in the abstract and texts is a bit exaggerated as 60% and 100%. I suggest that is made clear: 2/3 (60 percent) of co-morbid epilepsy and 12/12 (100 percent) of co-morbid migraine headache were undiagnosed.

We addressed these concerns accordingly. we revised as “We found that a considerably higher proportion of people with SPD had undiagnosed overall neurological disorder (87.5%; 14/16), epilepsy (60%; 3/5), as well as migraine headaches (100%; 12/12). (See abstract section page 2 line 50-51). 

Reviewer #2

General comments

Over all, study was done in an important and un addressed area and authors also reported good findings. But major concern what authors should consider is that, most of the patients with severe mental illness have manifestations which are really similar symptoms of neurologic disorders, so how could authors really overcome with his issue and in my suggestion most of the reports you found might be primary symptoms of SPD not neurologic disorders/what are the contents of the tool you used to measure Neurologic disorders like epilepsy in terms commonly reported neurologic like symptoms of SPD?

Section comments.

Thank you for your kind words. We appreciate that. 

Regarding the identification of the symptoms of epilepsy and migraine headache, we have used diagnostic criteria, which requires trained professionals who correctly differentiate the symptoms of those conditions from other psychiatric disorders (SPD). We have recruited trained data collectors who have adequate knowledge and experience in both the above neurologic disorders and SPDs (MSC psychiatry professionals). 

We have also clarified this issue in the revised version. We included the following statement in the method section “According to the ILAE guideline, an epilepsy case is defined as someone with an active, recurrent (two or more) condition of an epileptic seizure, which was unprovoked by an immediate cause. The same classification and definition of epilepsy have been used in previous studies in Ethiopia. (See method section page 5-6 line 146-149). 

Q2. Abstract: your back ground statement and finding are not consistent, I think your focus of study is Epilepsy and migraine headache, but you reported the prevalence of overall neurologic disorder. What was the tool you used to assess over all neurologic disorder, since neurologic disorder includes plenty of disorders? Please clarify what is meaning of overall neurologic disorder? Your conclusion is not also consistent with your title, what is importance of saying epidemiology of diagnosed and undiagnosed neurologic disorders? Since you only reported the finding of un diagnosed neurologic disorders in your result section of abstract.

You also reported that “Higher disability score (WHODAS score) was associated with increased odds of having neurological disorders” what kind of disability is it? We expect all most all of patients with severe mental disorders have disability, so how can you compare patients with SPD as disable and able? If so in terms of which domain of disability functioning?

We addressed these comments accordingly. We have included the following sentence in the result or conclusion section “The diagnoses rates of those disorders were significantly low, perhaps surprisingly zero for migraine headaches.” ). (See abstract section page 2 -3 line 59-60). 

We also included the following statement in method sections:

Definitions of terms

In the present study, overall neurological disorders represent the presence of either epilepsy and migraine headaches in patients with SPD. Research studies suggest that epilepsy and migraine headache are the two commonest chronic neurological disorders with episodic manifestations. (See method section page 6 line 170-174). 

Regarding disability, as the reviewer mentioned, epidemiological evidence suggests that a considerable proportion of people with SPD including schizophrenia, bipolar, and major depression have disabilities. For example, a study by Chaudhury and colleagues revealed that roughly one-third of patients with major depression (30%) and bipolar disorder (33.3%) and nearly two-thirds of patients with schizophrenia had a disability (64.3%. In the present study, we have included disability score (WHODAS score) as only as one of the factors contributing to the higher rates of neurologic disorders among patients with SPD as suggested by previous studies. WHODAS is not prepared to classify patients into disable or not disable. However, it is a continuous scale whereby higher scores indicating higher disabilities and lower scores indicating less disability. Therefore, we have used the total scores and we found higher scores suggesting increased risks of neurological conditions. Also, as the reviewer suggested further studies might be needed to test the mechanisms and the particular type of disabilities that are more linked with the increased risks. 

Q3,.Background:

A. Some modification in grammar, punctuation and English language should be done, for example line number 13 of background section; “Scientific evidence shows that early identification and treatment of comorbid medication conditions in patients with severe psychiatric disorders…….” Should be modified.

Line number 15 “improved” in should be modified as improvement in……

Done.

B. Even though, you reported scarcity of reports regarding prevalence of neurologic disorders among patients with severe mental illness it is better to include different magnitudes and burdens of neurologic disorders particularly epilepsy among patients with mental illness since there are literatures too.

We have addressed this issue accordingly. We included the following statement regarding the prevalence of migraine headache among patients with mental disorders (bipolar disorders) ‘” For example, a recent meta-analysis that assessed the prevalence of migraine headaches among patients with bipolar disorder involving seven studies on the subject found that roughly one-third of patients with bipolar disorder had comorbid migraine headaches (30.36%). (See introduction section page 3 line 76-79). 

Surprisingly, we did not find previous studies that determined the prevalence of epilepsy among patients with any mental illness. 

Q4,Method:

a. Study was conducted on 309……. it is better to be modified as study was conducted among 309…..

Inclusion criteria were…… modify

Done.

b. also Exclusion criteria: Your exclusion criteria was patients having severe illness at time of data collection affecting communication, we expect most of patients are with severe illness as name indicates SPD and communication problem is also one of the hallmark symptom of severe mental illness, of so how could you managed this issue.

Revised accordingly. See below revisions made concerning the eligibility criteria in the method section:

Inclusion and exclusion criteria

Participants satisfying the following criteria were included in this study: (1) adult (aged 18 years and above); (2) those patients who had a diagnosis SPD (including schizophrenia, bipolar, schizoaffective, and depressive disorders). Those patients who had a severe illness at the time of data collection (affecting communications) were excluded. (See method section page 5 line 118-122). 

c. Sampling procedure: It is good to using systematic random sampling, but my concern is how you managed or calculated k value for more than one treatment units? Since it is expected that more than one OPDs might be available in hospital?

As the reviewer mentioned there are around eight OPDs particularly allocated for the diagnoses and treatment of mood and psychotic disorders (severe mental disorders). However, there is one central area to collect and distribute charts of patients with the above diagnoses for those allocated OPDs whereby we selected our participants based on the calculated intervals. 

d. Measures: What was the importance of using structured clinical interviews for DSM- ΙѴ-TR axis Ι disorders (SCID) since you included already diagnosed patients. 

We have used SCID for two main reasons: First, to confirm the diagnoses because studies have suggested that a significant proportion of patients with mental disorders are misdiagnosed (>50%) especially in low and middle-income countries where non-psychiatry professionals are involved in diagnoses and treatment of mental disorders. Second, this paper is part of the project, and the other section of the projects deals with rates of misdiagnoses of severe mental disorders in specialized psychiatric centers in Ethiopia which requires application of SCID. 

Q5. Result: 

A. More than one-third of the participants had attended secondary school approximately half of the participants were Orthodox Christians…it needs coma in between school and approximately.

Done.

B. What was the prevalence of diagnosed neurologic disorders, since I cannot accesses within your manuscript?

We addressed this comment accordingly. We have included the following sentence in the result section “On the other hand, in this study, 12.5%, 40% and 0% of patients with overall neurologic disorder, epilepsy, and migraine headache respectively were diagnosed by the professionals.” (See result section page 8 line 230-233).

C. Factors associated with undiagnosed neurological disorders in patients with SPD

You found only higher WHODAS score was only variable associated with prevalence of neurologic disorders, would you explain your variable selection method and what was reason behind for this?

We have selected a variable base on (1): P-value; those variables with P-value <0.2 in the bivariate logistic regression model were included in multivariable analysis modeL; (2) we also considered Bradford Hill’s causality and association criteria in addition to the P-value.

D. Why don’t authors perform regression separately for epilepsy and migraine headache?

We did not perform regression analysis specifically for epilepsy and migraine headche due to small samples for the specific conditions. 

Q6. Discussion: 

A. Our prevalence was more than 2 times higher than reported prevalence of migraine headache in the general population as reported by the global burden of disease (GBD) study in 2016 (1.8%)……. Modify this sentence

Revised as “The prevalence of migraine headache in the current study was more than 2 times higher than the reported prevalence in the general population, according to the global burden of disease study in 2016 (1.8%). (See discussion section page 10 line 286-288).

B. Even though you did not compare with similar studies, your discussion is well written. 

Many thanks. 

C. Conclusion: 

D. Future longitudinal studies are needed to identify the factors associated with undiagnosed neurological disorders (what was the difficulty you faced to do this study design)?

Two main limitations if the current study: (1) the nature of the study prevented us to draw causal inference (cross-sectional study); (2) the small sample size particularly for the specific diagnosis. In fact, we did not analyze the factors associated with specific disorders because of the small sample size. 

We clarified as “Future longitudinal studies with adequate sample size are needed to identify the factors associated with undiagnosed neurological disorders particularly focusing on the specific disorders. (See conclusion section page 13 line 364-366).

E. Table 3: I strongly recommend you to write cross tabulation results for each variable.

 Done 

Reviewer #3

Q1. On sample size calculation, it is not clear that there are many comorbidities with severe psychiatric disorders. Therefore, better to deal with the prevalence of epilepsy and migraine headache. Report that you have considered the maximum sample size.

As the reviewer said, it is advisable to calculate the sample size for the specific disorders and then take the larger samples for the final analysis. However, as mentioned in the method section this study is part of a comorbidity survey among patients with severe psychiatric disorders in a specialized mental health setting in central Ethiopia which is aimed in determining the detection rates of the severe mental disorders, the prevalence of undiagnosed and diagnosed medical comorbidities, as well as the outcomes of treatments. For the project, the sample size has been calculated considering the prevalence of overall comorbid physical conditions among people with SPD. 

We included the following sentence in the limitation sections” Thirdly, the small number of participants (sample) should be considered in interpreting the results especially for the estimates of the specific disorders such as epilepsy and migraine headache. Also, future studies addressing this limitation are warranted.” (See discussion section page 12 line 354-356).

Q2. On statistical analysis

The researcher mentioned the independent variables; just remove from the statistical analysis. 

Removed. 

Q3. Result

a. Sociodemographic characteristics of the participants

Rather than mentioning as the following “Sociodemographic and clinical data for our participants were summarized in Table 1” it is better to write (table 1) at the end of the paragraph.

Done. 

b. The prevalence of undiagnosed neurological disorders in patients with SPD

Don’t put table 2 in each paragraph. Just put it at the end of last paragraph.

Done.

Q4. Discussion 

a. the researchers should at least search for additional literature that has been done from different data bases. If no they can compare their finding which is cross-sectional study with the meta-analysis.

Some of the justification for the difference in the finding are not scientifically sound according to the context that they are studying. Which the stating as follows

“There are wide ranges of explanations for higher prevalence rates of epilepsy among people with severe psychiatric disorders as compared with the reported prevalence in the general population. One of the possible explanations is that these disorders might have shared genetic factors”

Therefore, better to explain in more acceptable and in details so that the reader will be convinced from the finding.

As the reviewer mentioned, there are no studies on the prevalence as well as rates of undiagnosed and diagnosed chronic neurologic disorders with episodic manifestations such as epilepsy and migraine headache among patients with SPD. Hence, the vast majority of the comparisons of the findings of the current studies are with the findings from the general community. Now we have more elaborated or revised the discussions.

Our explanations regarding the higher rates of epilepsy and migraine headache are just to highlight or suggest the readers look into the details of those issues via various mechanisms. We also removed sentences regarding the prevalence in the community as they are not our objectives and not investigated in our study, as suggested by the reviewer.

---

## [Decision Letter · Decision Letter 1]

19 Oct 2020

Prevalence and correlates of diagnosed and undiagnosed epilepsy and migraine headache among people with Severe Psychiatric Disorders in Ethiopia

PONE-D-20-11101R1

Dear Dr. Ayano,

We’re pleased to inform you that your manuscript has been judged scientifically suitable for publication and will be formally accepted for publication once it meets all outstanding technical requirements.

Kind regards,

Kyoung-Sae Na, M.D.

Academic Editor

PLOS ONE

Additional Editor Comments (optional):

Reviewers' comments:

Reviewer's Responses to Questions

**Comments to the Author**

1. If the authors have adequately addressed your comments raised in a previous round of review and you feel that this manuscript is now acceptable for publication, you may indicate that here to bypass the “Comments to the Author” section, enter your conflict of interest statement in the “Confidential to Editor” section, and submit your "Accept" recommendation.

Reviewer #1: All comments have been addressed

Reviewer #2: All comments have been addressed

Reviewer #3: All comments have been addressed

2. Is the manuscript technically sound, and do the data support the conclusions?

Reviewer #1: Yes

Reviewer #2: Yes

Reviewer #3: Yes

3. Has the statistical analysis been performed appropriately and rigorously? 

Reviewer #1: Yes

Reviewer #2: Yes

Reviewer #3: Yes

4. Have the authors made all data underlying the findings in their manuscript fully available?

Reviewer #1: (No Response)

Reviewer #2: Yes

Reviewer #3: Yes

5. Is the manuscript presented in an intelligible fashion and written in standard English?

Reviewer #1: Yes

Reviewer #2: Yes

Reviewer #3: Yes

6. Review Comments to the Author

Reviewer #1: These minor editorial comments attached.

Dear Editor,

Thank you for forwarding the revised document. Authors have accommodated my comments and I have no big concern in terms of the content. But, I have a couple of points that can still need revision.

Ref # 18 is incomplete.

The article will benefit some English language editorial before it Is published. There are too many unnecessary parentheses and punctuation errors.

Thanks

Reviewer #2: Title: Prevalence and correlates of diagnosed and undiagnosed epilepsy and migraine headache among people with Severe Psychiatric Disorders in Ethiopia

Dear editor, thank you for giving chance of reading this manuscript again, I appreciate authors since they were highly responsive in their reaction to the review and significantly improved the quality of the manuscript. I suggest accepting this study for publication.

Good lack authors!!!

Reviewer #3: I have no significant concern for the time.I am happy that the authors addressed all the comments provided by me. I appreciate their effort.

7. PLOS authors have the option to publish the peer review history of their article (what does this mean?). If published, this will include your full peer review and any attached files.

Reviewer #1: No

Reviewer #2: **Yes: **Alemayehu Molla

Reviewer #3: **Yes: **Abraham Tamirat Gizaw

---

## [Editor Report · Acceptance letter]

26 Oct 2020

PONE-D-20-11101R1 

Prevalence and correlates of diagnosed and undiagnosed epilepsy and migraine headache among people with Severe Psychiatric Disorders in Ethiopia 

Dear Dr. Ayano:

I'm pleased to inform you that your manuscript has been deemed suitable for publication in PLOS ONE. Congratulations! Your manuscript is now with our production department. 

Kind regards, 

on behalf of

Dr. Kyoung-Sae Na 

Academic Editor

PLOS ONE